# Superoxide Dismutase-1 Intracellular Content in T Lymphocytes Associates with Increased Regulatory T Cell Level in Multiple Sclerosis Subjects Undergoing Immune-Modulating Treatment

**DOI:** 10.3390/antiox10121940

**Published:** 2021-12-03

**Authors:** Valentina Rubino, Anna Teresa Palatucci, Giuliana La Rosa, Angela Giovazzino, Francesco Aruta, Simona Damiano, Flavia Carriero, Mariarosaria Santillo, Rosa Iodice, Paolo Mondola, Giuseppina Ruggiero, Giuseppe Terrazzano

**Affiliations:** 1Dipartimento di Scienze Mediche Traslazionali, Università di Napoli “Federico II”, Via Pansini, 5, 80131 Napoli, Italy; valentina.rubino@unina.it (V.R.); angela.giovazzino@unina.it (A.G.); 2Dipartimento di Scienze, Università della Basilicata, Via dell’Ateneo Lucano, 85100 Potenza, Italy; annateresa.palatucci@unibas.it (A.T.P.); flavia.carriero@unibas.it (F.C.); giuseppe.terrazzano@unibas.it (G.T.); 3Dipartimento di Medicina Clinica e Chirurgia, Università di Napoli “Federico II”, Via Pansini, 5, 80131 Napoli, Italy; giuliana.larosa@unina.it (G.L.R.); simona.damiano@unina.it (S.D.); marsanti@unina.it (M.S.); 4Dipartimento di Neuroscienze, Scienze Riproduttive ed Odontostomatologiche, Università di Napoli “Federico II”, Via Pansini, 5, 80131 Napoli, Italy; francesco.aruta@unina.it (F.A.); rosa.iodice@unina.it (R.I.)

**Keywords:** multiple sclerosis, SOD-1, T lymphocytes, Treg, cytokines

## Abstract

Reactive oxygen species (ROS) participate in the T-cell activation processes. ROS-dependent regulatory networks are usually mediated by peroxides, which are more stable and able to freely migrate inside cells. Superoxide dismutase (SOD)-1 represents the major physiological intracellular source of peroxides. We found that antigen-dependent activation represents a triggering element for SOD-1 production and secretion by human T lymphocytes. A deranged T-cell proinflammatory response characterizes the pathogenesis of multiple sclerosis (MS). We previously observed a decreased SOD-1 intracellular content in leukocytes of MS individuals at diagnosis, with increasing amounts of such enzyme after interferon (IFN)-b 1b treatment. Here, we analyzed in depth SOD-1 intracellular content in T cells in a cohort of MS individuals undergoing immune-modulating treatment. Higher amounts of the enzyme were associated with increased availability of regulatory T cells (Treg) preferentially expressing Foxp3-exon 2 (Foxp3-E2), as described for effective Treg. In vitro administration of recombinant human SOD-1 to activated T cells, significantly increased their IL-17 production, while SOD-1 molecules lacking dismutase activity were unable to interfere with cytokine production by activated T cells in vitro. Furthermore, hydrogen peroxide addition was observed to mimic, in vitro, the SOD-1 effect on IL-17 production. These data add SOD-1 to the molecules involved in the molecular pathways contributing to re-shaping the T-cell cytokine profile and Treg differentiation.

## 1. Introduction

The ability of reactive oxygen species (ROS) to directly participate in the fine-tuning T-cell activation processes has been largely demonstrated [1]. The chemical reactivity, stability, and diffusion capability of each oxidant species represent a key element to regulate ROS-dependent signal transmission specificity. Moreover, ROS level and cell localization have been largely associated with the re-modelling of effector proteins and transcription factors in order to properly shape the complex orchestration of the immune response [2].

ROS-dependent regulatory networks are usually mediated by hydrogen peroxides, which are more stable and able to freely distribute inside cells [3]. SOD-1 physiologically mediates intracellular scavenging of oxygen radicals to H_2_O_2_ and molecular oxygen [4], thus representing the major source of peroxide generation inside the cells.

Growing evidence in vitro indicates that SOD-1 is secreted by several cell types, such as fibroblasts, hepatocytes, human neuroblastoma cells, Sertoli cells, and thymic epithelial cells [5,6,7,8]. SOD-1 secretion has been related to specific stress conditions by us and others [9,10].

Our recent data indicate that the antigen-dependent activation represents a triggering element for SOD-1 production and secretion by human T cells in vitro [11]. Indeed, increased amounts of SOD-1-containing micro-vesicles can be isolated from the supernatants of activated T cells [11]. The role of extracellular SOD-1 in shaping the antigen-dependent T-cell response needs to be investigated.

Orchestration of the immune response involves different T-cell subsets [12,13]; in this regard, pathogen control and tissue homeostasis appear to be finely regulated by the availability of memory versus naïve CD4 and CD8 populations, of natural (thymus-derived) and induced (microenvironment-dependent) regulatory T cells (Treg), as well as by differentiation and maintenance of particular T-cell cytokine profiles [14,15,16].

Immune-mediated mechanisms have been described to protect and damage neurons [17,18,19]. This paradoxical double role appears to specifically involve the T-cell subsets recruited into the neural microenvironment [19,20]. In this context, the generation/recruitment of Treg, preferentially expressing Foxp3-E2 and the induction of a non-inflammatory cytokine immune profile by activated T cells are believed to be crucial targets for therapeutic remodeling of the unbalanced immune response in MS [21,22]. A neuro-protective role of antioxidant enzymes has been also suggested [23,24].

A wide array of immune-modulating treatment options has been described for MS, but the molecular mechanisms underlying their ability to shape the deranged immune response in the central nervous system (CNS) is not completely understood [25,26,27]. In this regard, the main therapeutic approaches for MS are represented by: IFN β 1b, described to mediate attenuation of immune cell functions inside the CNS; glatimer acetate, a mixture of polypeptides able to exert immune suppression; teriflunomide, an inhibitor of dihydroorotate-dehydrogenase, a key mitochondrial enzyme involved in the de novo synthesis of pyrimidines in rapidly proliferating cells such as lymphocytes; dimethyl fumarate, able to mediate T_H_1/T_H_2 shift as well as to act on endogenous cellular antioxidative pathways involving the Nrf2 transcription factor; cladribine, a purine analogue acting as immune suppressor; and fingolimod, a structural analogue of sphingosine which is able to modulate sphingosine 1 phosphate receptor in immune and brain cells [25,26,27].

Previously, we described that SOD-1 is involved in T-cell receptor (TCR) signaling cascades relevant to fine-tuning the antigen-dependent T-cell response [11]. We also observed that SOD-1 concentration in cerebrospinal fluid (CSF) at the onset of MS is significantly reduced when compared to subjects suffering from other neurological diseases [28]. Furthermore, we observed the SOD-1 reduction in human peripheral blood mononuclear cells (PBMC) of MS subjects at diagnosis [28]. Finally, we found that IFN β 1b therapy increases the intracellular SOD-1 protein as well as its mRNA levels in PBMC of MS subjects [28].

Here, we investigate the SOD-1 intracellular content of T cells in a cohort of MS individuals undergoing immune-modulating treatment. Moreover, Treg lymphocytes, largely associated with the control of active disease [29,30], as well as the cytokine profile and the role of SOD-1 secretion in the regulation of inflammatory response were analyzed.

## 2. Patients and Methods

### 2.1. Patients

The study was carried out on 78 patients with a diagnosis of relapsing remitting multiple sclerosis (RR-MS; F = 51 (65%); mean age = 46.5 ± 9.2; EDSS = 2.1 ± 1.2) according to McDonald Criteria [31]. Similar features were observed when the MS-RR subjects were categorized according to the immune-modulating treatment they underwent (Appendix A). The enrolment of people with MS and their clinical management were performed at the Centre of Neuro-Immunology of the Department of Neuroscience, Reproductive Sciences and Odontostomatology, University of Naples “Federico II”. Naïve subjects as well as patients with a diagnosis of primary or secondary progressive multiple sclerosis were excluded. No relapse or corticosteroid treatment were reported in the 30-day period before inclusion. Fourteen healthy blood donors, age/sex matched with the patients, were enrolled as controls. The Institutional Review Board of the Ethics Committee of the University of Naples “Federico II” approved the study (Protocol number: 33017). All procedures were performed in accordance with the Declaration of Helsinki, as revised in 2008. All patients and healthy controls signed their informed consent to the study.

### 2.2. Cells, Immunofluorescence, and Flow Cytometry Analysis

PBMC were isolated by centrifugation of peripheral blood on a Ficoll-Paque cushion (GE Healthcare, Uppsala, Sweden) gradient. To evaluate possible oscillations in the results, two independent samples, obtained for each subject, were analyzed at a one-week interval and produced substantially comparable results (not shown). Immune fluorescence analysis was performed on PBMC by using FITC or Pe-Cy5 anti-human CD3 (BD Pharmingen, San Diego, CA, USA, clone UCHT1), FITC anti-human CD4 (BD Pharmingen, San Diego, CA, USA, clone RPA-T4), Pe-Cy7 anti-human CD8 (BD Pharmingen, San Diego, CA, USA, clone RPA-T8), Pe-Cy5 or Pe-Cy7 anti-human CD56 (BD Biosciences, Franklin Lakes, NJ, USA, clone NCAM16.2), PE anti-human Vα24 (Beckman Coulter, Brea, CA, USA, clone C15), FITC anti-human CD19 (eBioscience, San Diego, CA, USA, clone HIB19), PE anti-human CD25 (BD, Franklin Lakes, NJ, USA, clone M-A251), PE anti-human CD54 (BD, Franklin Lakes, NJ, USA, clone HA58), anti-SOD-1 mAb (Merk Life Science, Milan, Italy, clone IG6), FoxP3-all (eBioscience, San Diego, CA, USA, clone PCH101), FoxP3-exon 2 (eBioscience, San Diego, CA, USA, clone 250D/E4), FITC anti-human Ki-67 (BD, Franklin Lakes, NJ, USA, clone B56), FITC anti-human IFN-γ (eBioscience, San Diego, CA, USA, clone 4SB3), PE anti-human IL-17A (eBioscience, San Diego, CA, USA, clone eBio64DEC17), and PE anti-human IL-4 (BD, Franklin Lakes, NJ, USA, clone MP4-25D2). To analyze cytokine production PBMC were cultured overnight by using RPMI 1640 (Biochrom K.G., Berlin, Germany) supplemented with 10% heat-inactivated fetal bovine serum in the presence of phorbol 12-myristate 13-acetate (PMA, 50 ng/mL) and ionomycin (1 µg/mL) (Merk Life Science, Milan, Italy). To avoid extra-cellular cytokine export, cell cultures were incubated with 5 μg/mL of brefeldin-A (Sigma-Aldrich, St. Louis, MO, USA), as previously described [32]. When indicated, recombinant human SOD-1 (rhSOD-1 from Merk Life Science, Milan, Italy) or metal-depleted SOD-1 molecule (rhApo-SOD-1), both at 400 ng/mL [31], as well as, 50 µM hydrogen peroxide (H_2_O_2_), were added to the cultures. For intracellular detection the fixation/permeabilization solution kit BD Cytofix-Cytoperm (BD Biosciences, Franklin Lakes, NJ, USA) or the fixation and permeabilization FoxP3 buffer kit (eBioscence, San Diego, CA, USA), were employed according to the manufacturer’s instructions. For the comparative analysis of SOD-1 expression levels in T lymphocytes, fluorescence data were expressed as ratio of mean intensity fluorescence (MIF) value for the T-cell population and the control MIF value obtained after staining of the same cell subset with the isotype control mAb, as described [33]. Flow cytometry evaluation was performed by using the ATTUNE NxT acoustic focusing cytometer (Life Technologies, Carlsbad, CA, USA). Data analysis was performed by using FlowJo Software (FlowJo, LLC, Ashland, OR, USA).

### 2.3. Statistical Analysis

Statistical evaluation of data, by InStat 3.0 software (GraphPad Software Inc., San Diego, CA, USA), was performed by means of the Mann–Whitney test, Wilcoxon matched-paired test, or correlation assay, as indicated. Two-sided *p* values of less than 0.05 were considered to indicate statistical significance.

## 3. Results

### 3.1. T Cells from MS Individuals Undergoing Different Immune-Modulating Treatments, except Fingolimod, Show A Significant Increase of SOD-1 Intracellular Levels, as Compared with Healthy Controls

In order to investigate the ability of the immune-modulating treatment to affect the SOD-1 content in T lymphocytes of MS-RR individuals, we analyzed the intracellular content of such enzymes in circulating T cells from 78 MS-RR subjects, by using immunofluorescence and flow cytometry detection (see Patient and Method section). To preserve the physiological complexity of the immune subsets, the analysis was always been performed on PBMC.

As shown in Figure 1A, when we investigated the ability of different immune-modulating treatments, as represented by teriflunomide, glatiramer acetate, IFN β 1b, dimethyl fumarate, cladribine, and fingolimod, to affect the SOD-1 intracellular content of T lymphocytes, we found that T cells from MS-RR individuals, undergoing fingolimod treatment, revealed a significantly lower SOD-1 content, as compared with T lymphocytes from MS-RR subjects receiving the other immune-modulating therapies.

When the comparative analysis with healthy individuals was performed after stratification of the MS subjects, according to the immunomodulatory treatment they underwent (Figure 1B), we found a significant increase in the intracellular content of SOD-1 in T cells only in MS-RR subjects not treated with fingolimod (*p* < 0.005). We then compared the immune profile of MS-RR individuals treated with fingolimod, with MS-RR subjects undergoing teriflunomide, glatiramer acetate, IFN β 1b, dimethyl fumarate, or cladribine treatment. As shown (Figure 2), significant differences in the percentage (*p* < 0.0001) and number (*p* < 0.0001) of T cells, in the CD4/CD8 ratio (*p* < 0.0001; percentage of CD4^+^ T cells 58.88 ± 3.65 in fingolimod group vs. 73.88 ± 1.36 in the counterpart; *p* < 0.0001; percentage of CD8^+^ T cells 41.12 ± 3.65 in fingolimod group vs. 26.12 ± 1.36 in the counterpart; *p* < 0.0001), in the invariant natural-killer T-cell (iNKT) number (*p* < 0.005) and in the natural-killer (NK) percentage (*p* < 0.0005) were observed when MS-RR individuals treated with fingolimod were compared with MS-RR subjects receiving teriflunomide, glatiramer acetate, IFN β 1b, dimethyl fumarate, or cladribine therapy. No difference was observed in B lymphocyte number and percentage, nor in NK cell number and iNKT percentage between the groups.

Regulatory T cells (Treg) have been largely demonstrated to play a major role in the control of active MS disease [29,30]. Moreover, expression of exon 2 of Foxp3 transcription factor (Foxp3-E2) has been convincingly associated with effective regulatory activity of the Treg subset [22]. Thus, Treg level was evaluated in MS-RR individuals grouped according to the immune-modulating treatment administered; both the overall Foxp3 and the Foxp3-E2 expression were analyzed. We found (Figure 3A,B) that the percentage and number of the Treg subset were significantly higher in individuals with MS-RR undergoing all immunomodulatory treatments, except fingolimod as compared with MS-RR subjects treated with fingolimod (*p* < 0.005 and *p* < 0.0001, respectively). Foxp3-E2 expression analysis confirmed this observation (Figure 3C,D). Moreover, when comparative analysis of the Foxp3 isoform ratio was evaluated (Foxp3-E2/whole Foxp3 ratio) in the MS-RR cohort vs. healthy controls, an increased value was found in the MS-RR individuals, regardless the treatment they underwent (Figure 3E).

Therefore, immune-modulating treatments, except fingolimod, were associated with a significant increase of SOD-1 intracellular content in T lymphocytes (Figure 1), as well as with a higher number and percentage of the Treg subsets (Figure 3A–D). Moreover, an increased Foxp3-E2/whole Foxp3 ratio was observed in the MS-RR cohort undergoing immune-modulating treatment, regardless of the therapy administered.

### 3.2. Higher SOD-1 Intracellular Content in T Cells Associated with Increased Treg Level in Peripheral Blood of MS Individuals Undergoing Immune-Modulating Treatment, Except Fingolimod

The above data, significantly associated teriflunomide, glatiramer acetate, IFN β 1b, dimethyl fumarate, and cladribine treatment with an increased SOD-1 intracellular amount in T cells in MS-RR subjects, as compared with controls. A higher percentage of Treg characterized MS-RR subjects undergoing immune-modulating therapies, except fingolimod. In addition, in the above subjects (Figure 4) a significant positive correlation between intracellular SOD-1 content in T lymphocytes and Treg, as a whole (Figure 4A), or expressing Foxp3-E2 (Figure 4B), was observed.

To investigate whether SOD-1 intracellular amount in T lymphocytes per se might be associated with particular phenotype features in our MS-RR cohort, we analyzed the SOD-1 distribution in T cells of MS-RR subjects undergoing all immune-modulating treatments, except fingolimod (Figure 5A). As shown (Figure 5A), the intracellular amount of SOD-1 in T cells identified two sub-groups of MS-RR subjects—Group 1, with a SOD-1 level similar to healthy donors (MIF fold increase vs. isotype control <10) and Group 2, with a SOD-1 level higher than healthy donors (MIF fold increase vs. isotype control >10). Comparative analysis of the immune profile of Group 1 vs. Group 2 showed a significant increase (*p* < 0.05) in the number of T and NK lymphocytes (Figure 5C,G), as well as in the percentage of iNKT cells (Figure 5D).

Moreover, we observed a significant increase in both the percentage and number of the Treg subset in Group 2 as compared with Group 1 (Figure 6A,B; *p* < 0.05 and *p* < 0.005, respectively). The expression of Foxp3-E2 was found to strongly correlate with higher SOD-1 intracellular amount (Group 2) when both percentage (Figure 6C; *p* < 0.0005) and number (Figure 6D; *p* < 0.0001) of circulating Treg cells were evaluated. Foxp3-E2/whole Foxp3 ratio (Figure 6E) was higher in the MS-RR cohort than in healthy controls (*p* < 0.05). However, when the MS-RR subjects were analyzed according to their SOD-1 content in T cells (Group 1 vs. Group 2), a clear association of higher Foxp3-E2/whole Foxp3 ratio with SOD-1 amount in T lymphocytes was revealed (*p* < 0.005 Group 2 vs. controls; *p* < 0.001, when Group 1 and Group 2 individuals were compared).

Compelling evidence indicates that higher growth ability in vivo characterized the Treg subset when compared with conventional T cells [34]. Thus, we evaluated the growth ability of Treg and conventional CD4^+^ T lymphocytes (T_conv_) by analyzing their expression of the ki67 molecule, largely associated with cell proliferation [35].

As shown (Figure 7A), higher intracellular SOD-1 content in T cells was associated with increased proliferation ability of Treg (*p* < 0.05 when considering Group 2 individuals versus controls); at variance, (Figure 7B) increased ki67 expression was observed in all the MS-RR individuals when T_conv_ populations were analyzed in Group 1 and Group 2 subjects in comparison with controls (*p* < 0.005 and *p* < 0.0001, respectively).

Therefore, higher SOD-1 content in T lymphocytes was significantly associated with an increased percentage and number of Treg showing Foxp3-E2 expression in our cohort of MS-RR subjects. Moreover, a higher growing ability of the Treg subset was significantly associated with higher SOD-1 intracellular content in T lymphocytes.

### 3.3. SOD-1 Intracellular Content in T Cells and Cytokine Profile of MS-RR Individuals Undergoing Immune-Modulating Treatment, Except Fingolimod

Pro-inflammatory cytokine profile was largely associated with MS-RR clinical outcome [36]. Therefore, we analyzed the in vitro production of IFN-γ, IL-17, and IL-4 by T cells obtained from MS-RR Group 1 and Group 2 individuals, classified according to their intracellular SOD-1 content in T lymphocytes.

As shown, (Figure 8) no difference was observed in proinflammatory cytokine production between Group 1 and Group 2 MS-RR individuals and healthy controls (Figure 8A,B). At variance, significant difference in the ability to produce IL-4 was revealed when Group 1 and Group 2 subjects were compared (Figure 8C; *p* < 0.0001). In addition, IL-4 production was significantly lower than controls in Group 1 individuals, showing the lowest intracellular amount of SOD-1 in T cells (Figure 8C; *p* < 0.0005). Thus, MS-RR individuals undergoing teriflunomide, glatiramer acetate, IFN β 1b, dimethyl fumarate, and cladribine treatment showed a significant correlation between SOD-1 intracellular content in T cells and IL-4 production.

### 3.4. SOD-1 Addition to Activated T Cells Significantly Affected Their IL-17 Production In Vitro

Our previous results demonstrated that T-cell activation is associated with increased SOD-1 production accompanied by extracellular export of the enzyme by micro vesicle secretion [10,11].

To investigate the role of extracellular SOD-1 in T-cell response, we analyzed the effect of the in vitro addition of SOD-1 on cytokine production in T cells isolated from healthy donors. Since the proinflammatory cytokine profile has been largely related to MS activity in human and mouse models [36] we focused on the T_H_1, T_H_2, and T_H_17 profile. As shown in Figure 9, the supplementation of rhSOD-1 to activated T cells increased IL-17 production (*p* < 0.0005) by CD4^+^ T-cell effectors (Figure 9A), without notable effects on IFN-γ production by both CD4^+^ and CD8^+^ T lymphocytes (Figure 9B,C).

This effect seemed to depend on the enzymatic activity of SOD-1, since it disappeared when rhApo-SOD-1, the metal-depleted enzyme with no catalytic dismutase activity [37], was added to activated T cells (Figure 9A). In order to investigate the ability of hydrogen peroxide, mainly produced by SOD-1 molecules, to directly affect T-cell cytokine production, we analyzed, in vitro, the effect of H_2_O_2_ addition on IL-17 production by activated T cells. As shown in Figure 9D, the addition of H_2_O_2_ to activated T-cell cultures significantly increased the percentage of IL-17-producing CD4^+^ T lymphocytes (*p* < 0.05). This effect was observed to be preferentially mediated by hydrogen peroxide addition in the last 4 h of activation. No significant effects were observed on IFN-g production by T lymphocytes (not shown).

## 4. Discussion

SOD-1 induction as well as extracellular secretion by activated T cells has been previously described [11]. Here we showed, in a cohort of 78 MS-RR individuals undergoing immune-modulating treatment, a correlation between the intracellular content of SOD-1 in T cells and the increasing presence of circulating Treg expressing the exon 2 of Foxp3 molecule, largely associated with effective Treg immunomodulatory properties [21,22]. Moreover, the ability of secreted SOD-1 to participate in a proinflammatory loop has been suggested by the effect of SOD-1 addition in vitro on IL-17 production by T-cell effectors from healthy donors. Hydrogen peroxide is able to mimic such effect. These observations add SOD-1-dependent pathways to the molecular networks involved in the regulation of both T-cell proinflammatory cytokine profile and Treg immune-modulation.

The involvement of ROS in ensuring optimal antigen-dependent T-cell response has been largely described [1,2]; moreover, the key role of peroxides and consequently of SOD-1, the enzyme physiologically responsible for intracellular peroxide generation, has also been revealed [3,4]. We previously found that TCR triggering is associated with rapid TCR/SOD-1 co-localization, SOD-1 mRNA induction, and active SOD-1 microvesicle secretion by activated T lymphocytes [11].

MS, as well as other neurodegenerative pathologies, is characterized by deranged inflammatory responses [18,38]. A number of immune-modulating treatments are currently employed to effectively control the disease. Here, we showed that all immune-modulating treatments, except fingolimod, are able to induce increasing SOD-1 intracellular levels in T cells from MS-RR individuals.

Intracellular SOD-1 content in T cells is dependent on both SOD-1 induction and secretion by activated T lymphocytes [11]. Thus, an altered balance between intracellular and secreted SOD-1 molecule in pathological T-cell effectors might participate in the deranged activation of T-cell clones, largely described in MS [18]. The decreased SOD-1 content in leukocytes, that we previously described [28] in MS at disease onset, might be related to a higher secretion of the enzyme able to enhance pro-inflammatory pathways. At variance, the increasing amount of SOD-1 in T cells, that we observed in MS subjects undergoing immune-modulating treatments, might be associated with a decreased SOD-1 secretion by modulated immune effectors.

Low intracellular SOD-1 level in T lymphocytes characterized MS-RR subjects treated with fingolimod (Figure 1). Such a feature is conceivable with the ability of the drug to trap lymphocytes inside secondary lymphoid organs [25,26,27], likely inhibiting SOD-1 induction, usually associated with antigen-dependent T-cell activation [11]. This mechanism, likely hampering the recognition of neural antigens by pathological T cells, might underlie the lack of SOD-1 increase that we observed in MS subjects undergoing effective fingolimod treatment.

Compelling evidence has consistently associated the percentage of circulating Treg subset, specifically expressing Foxp3-E2 with effective immune modulation in MS [29,30]. In this study we observed a clear increase of SOD-1 intracellular content in T cells of MS-RR individuals undergoing all immune-modulating therapies, except fingolimod. In this last patient group, significant positive correlation between intracellular SOD-1 content in T cells and circulating regulatory T cells, was observed (Figure 4). Comparison of the immune profile of MS-RR subjects categorized according to the therapy they undergo (fingolimod vs. all treatment except fingolimod) revealed significant changes in circulating T-cell number and percentage, CD4/CD8 ratio, iNKT, B cell number, and NK percentage (Figure 2). Some of these changes (i.e., T-cell percentage and CD4/CD8 ratio) were not maintained when subjects were specifically grouped according to their SOD-1 amount in T lymphocytes. At variance, the significant increase of circulating Treg, specifically expressing Foxp3-E2, was consistently associated with intracellular SOD-1 level in T lymphocytes. This is conceivable with the idea that SOD-1-dependent intracellular pathways may be relevant for remodeling Treg differentiation and activity in individuals with MS-RR. In this sense, we observed that the intracellular content of SOD-1 in T cells was associated with increased Treg growth ability when Group 2 MS-RR subjects were compared with controls (Figure 6A).

SOD-1 represents a major target of the mTOR enzyme [39], whose activation has been largely associated with the differentiation and expansion of the Treg subset [21]. Indeed, reversible mTOR-dependent SOD-1 phosphorylation has been described to mediate SOD-1 inhibition [40,41]. Our data strongly associate SOD-1 intracellular amounts in T cells with increased Treg differentiation/activity. These observations depict a complex scenario in which SOD-1/mTOR intracellular interplay might finely tune inflammatory pathways in the context of an immune-mediated disease. The mechanisms underlying SOD-1/mTOR cross talk in T-cell regulation need further investigation.

Immune-mediated diseases are usually associated with a deranged pro-inflammatory activity [36]. Moreover, remodeling of the cytokine profile has been largely correlated with effective immune modulation. The possibility that redox-dependent pathways can contribute to the molecular network ensuring a balanced immune profile must be adequately investigated. The observation that rhApo-SOD-1, the metal-depleted enzyme with no catalytic dismutase activity [37], is unable to interfere with the production of cytokines by activated T cells, points to a key role for peroxide in the SOD-1-dependent immune-regulatory pathways. Indeed, H_2_O_2_-mediated activity has been observed to inhibit protein tyrosine phosphatases (PTP), thus increasing protein tyrosine kinase phosphorylation [42,43]. The observed ability of hydrogen peroxide to increase in vitro IL-17 production by activated T cells, when preferentially added in the last four hours of culture, strongly supports such hypothesis.

Here, we describe a significant increase of SOD-1 intracellular content in T cells of MS-RR individuals undergoing immune-modulating therapies, while reduced SOD-1 levels were previously [28] observed to characterize MS-RR subjects at disease onset. Moreover, we demonstrated that extracellular SOD-1 may contribute to boost the proinflammatory cytokine profile in vitro. These data are conceivable with the hypothesis that effective immunomodulatory treatments might decrease the proinflammatory activity of T cells also reducing the secretion of SOD-1 micro vesicles by pathological T lymphocytes [11,28]. The quantitative changes in SOD-1 intracellular content that we observed in MS-RR individuals at diagnosis and after immune-modulation strongly support this idea.

## 5. Conclusions

Immune tolerance is a complex phenomenon involving a number of molecular networks, as well as multiple oxidative pathways. All our data show that SOD-1 is involved in the biological pathways contributing to re-shaping the T-cell cytokine profile and Treg differentiation, thus proposing new molecular tools for innovative immune-modulating strategies. The possibility that an altered mTOR/SOD-1 intracellular ratio might be associated with the deranged tolerance control, likely relevant for MS-RR pathogenesis, needs further investigation.

## Figures and Tables

**Figure 1 antioxidants-10-01940-f001:**
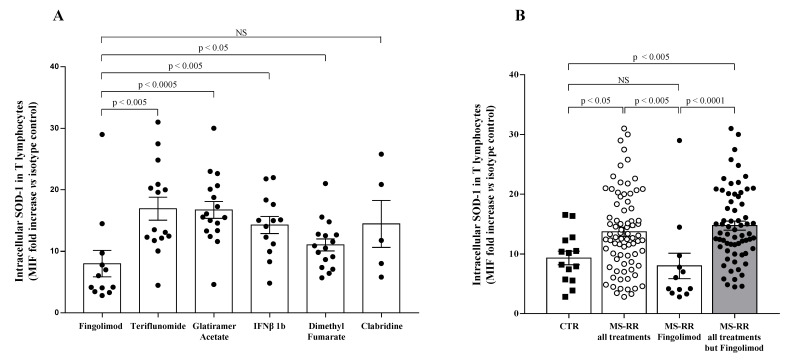
T cells from MS-RR individuals undergoing immune-modulating treatment, except fingolimod, showed significant increase of SOD-1 intracellular levels, as compared with healthy controls. (**A**) SOD-1 level in T lymphocytes obtained from MS-RR subjects undergoing different immune-modulating treatments. SOD-1 intracellular amount was expressed as the ratio of the mean intensity fluorescence (MIF) value for T-cell population and the control MIF value obtained after staining of the same cell subset with the isotype control mAb (see Patient and Method section). Each column refers data obtained in MS-RR subjects categorized according to the treatment they underwent. As shown, all treatments were able to increase SOD-1 levels in T lymphocytes, as compared with fingolimod. (**B**) Intracellular SOD-1 content in T lymphocytes of healthy controls (*n* = 14; white column with black squares), MS-RR subjects as a whole (*n* = 78; white column with white dots), MS-RR individuals undergoing fingolimod treatment (*n* = 12; white column with black dots), and MS-RR individuals undergoing all treatments, except Fingolimod (*n* = 64; grey column with black dots). As shown, when MS-RR individuals were grouped according to the treatment, significant SOD-1 increase in T cells only characterized the subjects undergoing all immune-modulating therapies, except fingolimod. Statistical evaluation of data was performed by means of the Mann–Whitney test. Statistically significant values are indicated.

**Figure 2 antioxidants-10-01940-f002:**
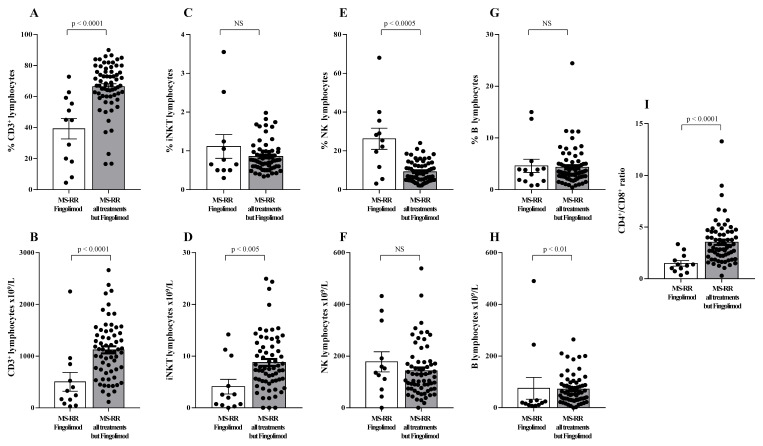
Immune profile of MS-RR individuals grouped according to the administered treatment. White column with black dots represents the MS-RR individuals undergoing fingolimod therapy; grey column with black dots indicates the MS-RR subjects undergoing all treatments, except fingolimod. (**A**,**C**,**E**,**G**) Percentage of circulating CD3^+^, iNKT, NK, and B lymphocytes, respectively. (**B**,**D**,**F**,**H**) Number of circulating CD3^+^, iNKT, NK, and B lymphocytes, respectively. (**I**) CD4^+^/CD8^+^ ratio. Statistical evaluation of data was performed by means of the Mann–Whitney test.

**Figure 3 antioxidants-10-01940-f003:**
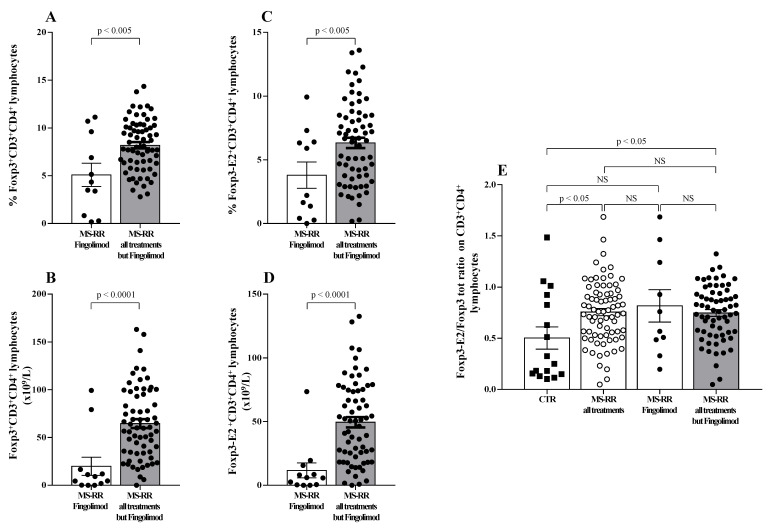
MS-RR individuals undergoing immune-modulating treatments, except fingolimod showed a Treg percentage and number significantly higher than the controls. White column with black dots represents the MS-RR individuals undergoing fingolimod therapy and grey column with black dots indicates the MS-RR subjects undergoing all treatments, except fingolimod. (**A**,**B**) Percentage and number of Treg cells, evaluated by Foxp3 expression in CD4^+^ lymphocytes. (**C**,**D**) Percentage and number of Treg cells evaluated by Foxp3-E2 expression in CD4^+^ T lymphocytes. (**E**) Comparative analysis of Foxp3-E2/whole Foxp3 ratio. Statistical evaluation of data was performed by means of the Mann–Whitney test.

**Figure 4 antioxidants-10-01940-f004:**
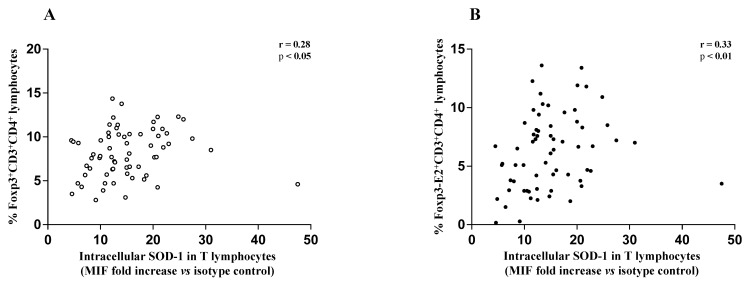
(**A**) Scatter plot showing positive correlation between the frequency of circulating Foxp3-expressing CD4^+^ T lymphocytes and intracellular SOD-1 content in T cells of MS-RR individuals undergoing immune-modulating treatments, except fingolimod; (r = 0.28; 95% confidence interval (CI) = 0.03 to 0.49 by two-tailed Spearman correlation *p* < 0.05). (**B**) Scatter plot showing positive correlation between the frequency of circulating Foxp3-E2-expressing CD4^+^ T lymphocytes and intracellular SOD-1 content in T cells of MS-RR individuals undergoing immune-modulating treatments, except fingolimod; (r = 0.33; 95% CI = 0.091 to 0.54 by two-tailed Spearman correlation; *p* < 0.01).

**Figure 5 antioxidants-10-01940-f005:**
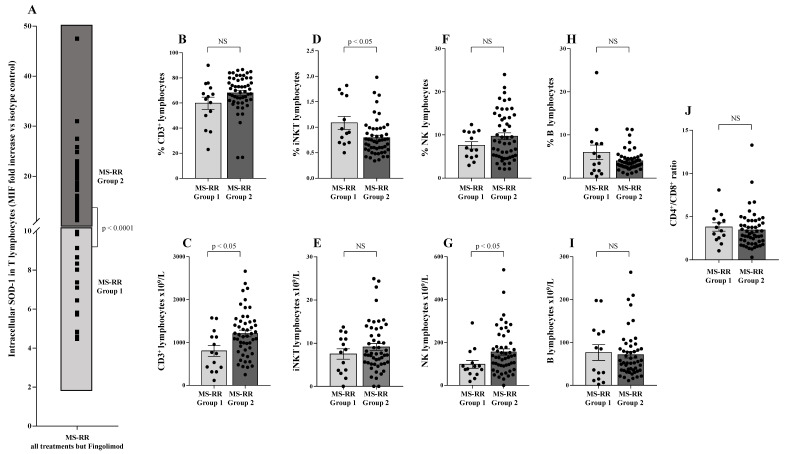
SOD-1 intracellular content in T cells identified two sub-groups of MS-RR individuals undergoing immune-modulating therapy, except fingolimod. (**A**) Analysis of SOD-1 distribution in T lymphocytes of MS-RR subjects undergoing all treatments, but fingolimod; as shown, Group 1 is MS-RR subjects characterized by SOD-1 intracellular content in T cells similar to healthy individuals (light grey box; SOD-1 MIF fold increase vs. isotype control <10); Group 2 is MS-RR subjects characterized by SOD-1 intracellular content in T cells significantly higher than healthy individuals (dark grey box; SOD-1 MIF fold increase vs. isotype control >10). (**B**,**D**,**F**,**H**) Percentage of circulating CD3^+^, iNKT, NK, and B lymphocytes, respectively. (**C**,**E**,**G**,**I**) Numbers of CD3^+^, iNKT, NK, and B lymphocytes, respectively. (**J**) CD4^+^/CD8^+^ ratio. Statistical evaluation of data, was performed by means of the Mann–Whitney test.

**Figure 6 antioxidants-10-01940-f006:**
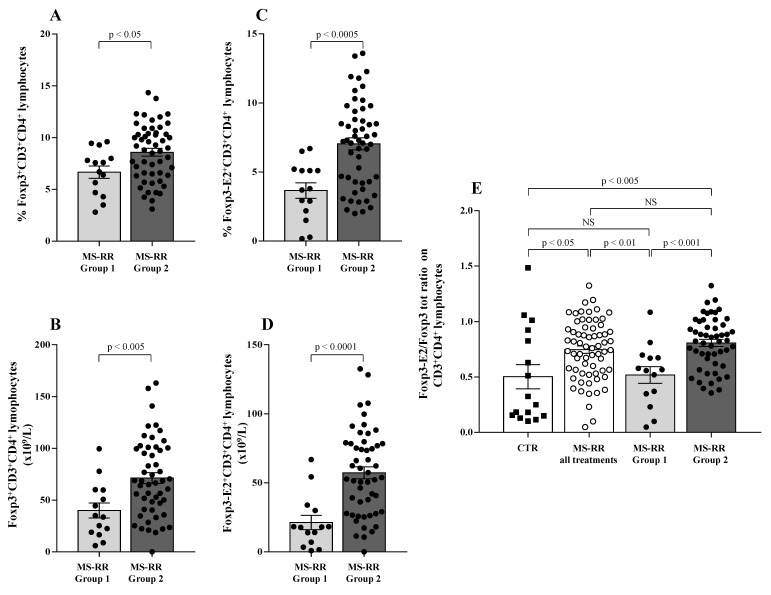
Higher SOD-1 content in circulating T cells of MS-RR individuals was associated with significant increase of Treg-expressing Foxp3-E2. Light grey column with black dots represents the MS-RR Group 1 subjects; dark grey column with black dots indicates the MS-RR Group 2 individuals. (**A**,**B**) Intranuclear Foxp3 transcription factor expression in CD4^+^ T lymphocytes. (**C**,**D**) Foxp3-E2 transcription factor expression in CD4^+^ T lymphocytes. (**E**) Comparative analysis of Foxp3-E2/whole Foxp3 ratio in the MS-RR cohort (white column with white dots) versus healthy controls (white column with black squares). Statistical evaluation of data was performed by means of the Mann–Whitney test.

**Figure 7 antioxidants-10-01940-f007:**
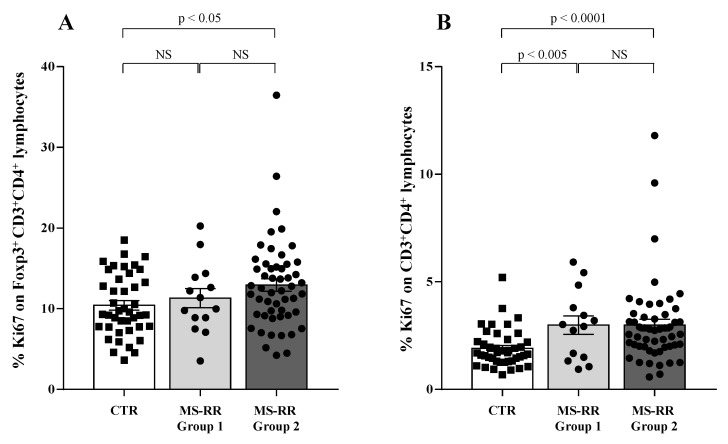
Higher intracellular SOD-1 content in T cells in MS-RR Group 2 subjects was associated with increased proliferation ability of Treg. White column with black squares shows the healthy controls; light grey column with black dots represents the MS-RR Group 1; dark grey column with black dots indicates the MS-RR Group 2 individuals. (**A**) ki67 expression, largely associated with cell proliferation, in the Treg subset. (**B**) ki67 expression in CD4^+^ T lymphocytes (T_conv_). Statistical evaluation of data was performed by means of the Mann–Whitney test.

**Figure 8 antioxidants-10-01940-f008:**
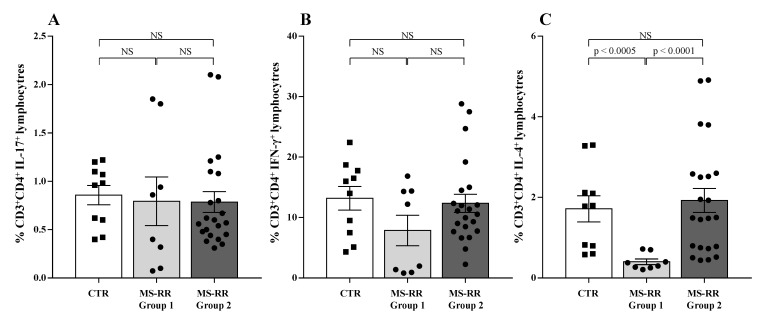
Low SOD-1 intracellular content in T cells correlated with reduced IL-4 production by CD4^+^ T lymphocytes in MS-RR individuals. To analyze the cytokine profile, PBMC were cultured overnight in the presence of PMA and ionomycin. White column with black squares represents the healthy controls; light grey column with black dots represents the MS-RR Group 1 subjects; dark grey column with black dots indicates the MS-RR Group 2 individuals. (**A**) IL-17 production by CD4^+^ T lymphocytes. (**B**) IFN-γ production by CD4^+^ T lymphocytes. (**C**) IL-4 production by CD4^+^ T lymphocytes. Statistical evaluation of data was performed by means of the Mann–Whitney test.

**Figure 9 antioxidants-10-01940-f009:**
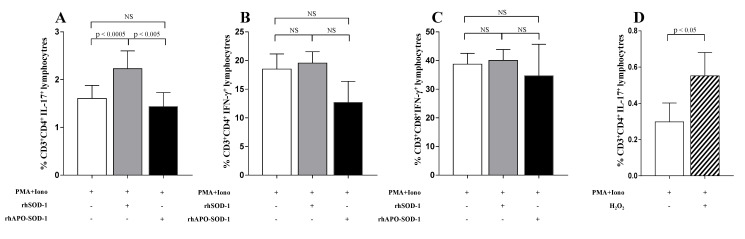
Addition of enzymatic active recombinant SOD-1 and H_2_O_2_ to activated T cells, significantly increased in vitro IL-17 production by CD4^+^ T lymphocytes. To evaluate the cytokine production, in vitro, an overnight culture of PBMC in the presence of PMA and ionomycin was performed. White columns indicate cultures with PMA and ionomycin alone; grey columns indicate cultures with recombinant human SOD-1 (rhSOD-1); black columns represent cultures with metal-depleted SOD-1 molecules (rhApo -SOD-1). (**A**) IL-17 production by CD4^+^ T lymphocytes. (**B**) IFN-γ production by CD4^+^ T lymphocytes. (**C**) IFN-γ production by CD8^+^ T lymphocytes. (**D**) IL-17 production by CD4^+^ T lymphocytes in the presence of PMA and ionomycin alone (white column) or after addition of H_2_O_2_ (50 mM) in the last 4 h of culture (striped column). Statistical evaluation of data was performed by means of the Wilcoxon matched-paired test.

## Data Availability

The data presented in this study are available in the article and Appendix A.

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
