# Peer review of "Superoxide Dismutase-1 Intracellular Content in T Lymphocytes Associates with Increased Regulatory T Cell Level in Multiple Sclerosis Subjects Undergoing Immune-Modulating Treatment"

_antioxidants, 2021, doi:10.3390/antiox10121940_

Round 1
Reviewer 1 Report
The manuscript by Rubino et al. describes that the intracellular content, Superoxide Dismutase-1 (SOD-1) in T lymphocytes is associated with increased regulatory T cells (Treg) in multiple sclerosis (MS) subjects undergoing immune-modulating treatment. This is an interesting work for the management of MS, but there are a few concerns:
- The quality of all the figures is bad. It's hazy and therefore difficult to read and understand.
- Could the authors show the relapse rate or Expanded Disability Status Scale (EDSS) score of the MS patients on different therapies?
- What is the level of SOD-1 in T lymphocytes of treatment naïve, patients not on any therapy, MS patients? The authors should include this data in figure 1 and then do the analysis.
- In figure 2, what was the percentage of CD4 and CD8 T cells?
- In Figures 2 and 3, the data on treatment naïve MS patients is missing.
- The description of Group 1 and Group 2 is very confusing.
- In figure 7, what was the level of IL-10 production?
- The authors show an increase of Tregs in groups with increased SOD-1 in the beginning. Later on, they show increased IL-17 production on the addition of SOD-1 in cultures. These are two opposing effects of SOD-1 in the same model. How do the authors speculate on this?
- The discussion is unnecessarily lengthy. The authors should be very specific and precise in their discussion.
- There are several grammatical errors. Therefore, I would recommend the authors to have a person whose native language is English, proofread the manuscript.
Author Response
Point by point reply
Reviewer 1
- The quality of all the figures is bad. It's hazy and therefore difficult to read and understand.
We apologize for the bad quality of the images. As requested, a higher resolution has been employed for all the figures embedded in the revised paper. In addition, each figure has been also attached as pdf file at higher resolution. Of course, a TIFF image will be provided afterwards.
- Could the authors show the relapse rate or Expanded Disability Status Scale (EDSS) score of the MS patients on different therapies?
As indicated in the Patients and Method section no relapse or corticosteroid treatment were reported in 30 days before inclusion in the study; our cohort was characterized by a F/M ratio = 51/27; a mean age = 46.5± 9.2; a EDSS = 2.1±1.2. Similar features were observed when the MS-RR subjects were categorized according with the immune modulating treatment they underwent. As requested, the data specifically concerning the MS-RR individuals grouped according their therapy have been included in the supplementary Table 1.
- What is the level of SOD-1 in T lymphocytes of treatment naïve, patients not on any therapy, MS patients? The authors should include this data in figure 1 and then do the analysis.
The study is specifically focused on the analysis of SOD1 intracellular content in T cells in MS-RR subjects undergoing immune modulating treatments. Thus, we not enrolled naïve MS-RR individuals; in the revised manuscript such condition has been specifically included in the enrollment criteria. We apologize for the misunderstanding.
- In figure 2, what was the percentage of CD4 and CD8 T cells?
As shown in Figure 2, we preferred CD4/CD8 ratio as an indicator of CD4 and CD8 T cell subset distribution in our cohort. As requested, the percentage of CD4 and CD8 T lymphocytes have been included in the result section of the revised manuscript.
- In Figures 2 and 3, the data on treatment naïve MS patients is missing.
As already indicated in the point 2, in our study the naïve MS-RR subjects were not enrolled.
- The description of Group 1 and Group 2 is very confusing.
We apologize for the not clear description. Compelling evidence consistently associated the percentage of circulating Treg subset, specifically expressing Foxp3E-2 with effective immune modulation in MS (Sospedra M. et al. Annu. Rev. Immunol., 2005; Talbot S. et al. Annu. Rev. Immunol., 2016). In our study, we observed a clear significant increase of SOD-1 intracellular content in T cells of MS-RR individuals undergoing all immune modulating therapies except Fingolimod. In this last patient group, a significant increase of the Treg subset, specifically expressing Foxp3-E2 was observed. In addition, significant positive correlation between intracellular SOD-1 content in T cells and circulating regulatory T cells, was observed (Fig. 4 revised manuscript). Thus, to deeply investigate the difference in immune profile, likely associated with different SOD-1 T cell intracellular amount, we categorized the MS-RR subjects according to their SOD-1 intracellular level in T cells. Indeed, Group 1 individuals were characterized by SOD-1 level similar to healthy controls, while Group 2 subjects showed SOD-1 intracellular content in T cells significantly higher than controls (SOD-1 content in Group 1 vs SOD-1 content in Group 2 p<0.0001) . As requested, the revised paper more clearly describes such issue.
- In figure 7, what was the level of IL-10 production?
We agree with the reviewer about the potential involvement of IL-10 production by the Tr1 subset in the deranged regulation of pro-inflammatory response in MS-RR pathogenesis. In this context, we only analyzed TH1, TH2 and TH17 profile in our cohort. Further studies will be performed to deeply study Tr1 role in MS-RR.
- The authors show an increase of Tregs in groups with increased SOD-1 in the beginning. Later on, they show increased IL-17 production on the addition of SOD-1 in These are two opposing effects of SOD-1 in the same model. How do the authors speculate on this?
Our previous studies showed the involvement of SOD-1 secretion in T cell activation (Terrazzano et al., Biochim. Biophys. Acta., 2014). To investigate on the role of SOD-1 export on cytokine production by T cells, we analyzed the effect of SOD-1 addition on cytokine profile of T cells from healthy donors. The results suggest the presence of a pro-inflammatory loop preferentially involving the extracellular, not the intracellular SOD-1. As requested, this aspect has been adequately clarified in the result section of the revised manuscript.
- The discussion is unnecessarily lengthy. The authors should be very specific and precise in their discussion.
As requested, in the revised paper the discussion has been modified according to the referee suggestions.
- There are several grammatical errors. Therefore, I would recommend the authors to have a person whose native language is English, proofread the manuscript.
As requested, the manuscript has been modified by a native English revisor.
Reviewer 2 Report
In the current manuscript, Rubino et. al., show that multiple sclerosis patients undergoing immune-modulating treatment (except the treatment Fingolimod) can be stratified into groups with high versus low intracellular SOD1 levels in T lymphocytes, and that patients with high levels of intracellular SOD1 in T cells also have a higher percentage of Tregs expressing Foxp3-E2. By adding human recombinant SOD1 or hydrogen peroxide to T cell cultures, an increase in IL-17 production was observed highlighting a role for SOD1 in determining the T cell cytokine profile and Treg differentiation. The results of this study are interesting, however the following issues need to be addressed before I can recommend this manuscript for publication.
- The text on all the graphs is very blurry, almost unreadable.
- The way figures are referred to in the text is inconsistent, for example, Panel A of Figure 1 and Fig. 1B.
- Abbreviations for some cell types, such as NK and iNKT should have the full name described first.
- Currently, both iNKT and NKTi are used as abbreviations.
- Figure 3 – ‘I’ should be ‘E’.
- Figure 4 – ‘L’ should be ‘K’.
- The results observed due to the stratification of patients into groups with high versus low intracellular SOD1 levels in T lymphocytes are very interesting. However, before the authors conclude that the variables are “correlated” with each other, shouldn’t statistical correlation tests be performed?
- It is not clear what the relevance of looking at the ‘immune profile’ of MS-RR individuals is (figure 2 and figure 4). There seems to be some interesting data but there is currently no discussion around the specific changes that are observed and what they might mean etc.
- Currently there is very minimal discussion around the potential reasons for the different results observed when patients were taking Fingolimod compared to the other treatments. Please expand this section and include a reference for the current sentence in the discussion.
- Do the authors have any thoughts on why some MS patients might have T cells with intracellular SOD1 levels that are similar to controls while others have much higher levels? What other factors might be contributing to this?
- The manuscript should be carefully checked for correct use for English. Incorrect word choice, gramma and sentence structure are noted throughout.
Author Response
Reviewer 2
- The text on all the graphs is very blurry, almost unreadable.
We apologize for the bad quality of the images. As requested, a higher resolution has been employed for all the figures embedded in the revised paper. In addition, each figure has been also attached as pdf file at higher resolution.
- The way figures are referred to in the text is inconsistent, for example, Panel A of Figure 1 and Fig. 1B.
We apologize for the inaccuracy. The revised paper has been modified according to the referee request
- Abbreviations for some cell types, such as NK and iNKT should have the full name described first. Currently, both iNKT and NKTi are used as abbreviations.
- Figure 3 – ‘I’ should be ‘E’.
- Figure 4 – ‘L’ should be ‘K’.
We apologize for the inaccuracies. The revised paper has been modified according to the referee request
- The results observed due to the stratification of patients into groups with high versus low intracellular SOD1 levels in T lymphocytes are very interesting. However, before the authors conclude that the variables are “correlated” with each other, shouldn’t statistical correlation tests be performed?
As requested, in the revised manuscript we specifically included the test (Fig.4) showing the positive correlation between intracellular SOD-1 content and circulating regulatory T cells.
- It is not clear what the relevance of looking at the ‘immune profile’ of MS-RR individuals is (figure 2 and figure 4). There seems to be some interesting data but there is currently no discussion around the specific changes that are observed and what they might mean etc.
We observed that increased SOD-1intracellular level in T cells seems to characterize immune modulating therapies except Fingolimod. Our data also show that some changes in immune profile observed in individuals undergoing all immune modulating treatments except Fingolimod (i.e. CD4/CD8 ratio) are not maintained when subjects were specifically grouped according with their SOD-1 amount in T lymphocytes. Interestingly, the significant increase of circulating Treg, specifically expressing Foxp3-E2, has been consistently observed also when individuals were specifically grouped according with their intracellular SOD-1 level in T lymphocytes. This observation is conceivable with the idea that SOD-1 dependent pathways might participate in Treg differentiation. Recent data (Tsang et al., Mol. Cell. Oncol. 2018) showing the ability of mTOR, a key Treg regulator, to control SOD-1 activity, strongly support such hypothesis. As requested, in the discussion section we discuss such hypotheses.
- Currently there is very minimal discussion around the potential reasons for the different results observed when patients were taking Fingolimod compared to the other treatments. Please expand this section and include a reference for the current sentence in the discussion.
Multiple mechanisms have been associated with the immune modulating properties of the different treatments, currently employed in clinical management of MS. Particularly, Fingolimod immune modulation has been associated with the block of lymphocytes in secondary lymphoid organs, thus hampering their ability to recognize and damage autologous molecular targets in peripheral tissues (Wingerchuk et al., Mayo Clin. Proc., 2014; De Kleijn et al., Int. J. Mol. Sci., 2020). This mechanism is conceivable with the possibility that this drug might be unable to activate the intracellular pathways usually involved in antigen recognition, as represented by SOD-1 induction (Wingerchuk et al., Mayo Clin. Proc., 2014; De Kleijn et al., Int. J. Mol. Sci., 2020). This condition has been proposed to underlie the lack of SOD-1 increase by us observed in MS subjects undergoing effective Fingolimod treatment. As requested, the discussion has been modified to more clearly discuss such issue.
- Do the authors have any thoughts on why some MS patients might have T cells with intracellular SOD1 levels that are similar to controls while others have much higher levels? What other factors might be contributing to this?
The immune tolerance is a complex phenomenon involving redundant molecular network, as well as multiple oxidative pathways. These conditions might underlie different immune modulating responses. Accordingly, the intracellular SOD-1 amount in immune effectors as well as the SOD-1 dependent oxidative pathways might consistently differ in single MS-RR subjects undergoing immune-modulating therapy. As requested, the revised discussion addresses such intriguing issue.
- The manuscript should be carefully checked for correct use for English. Incorrect word choice, gramma and sentence structure are noted throughout.
As requested, the manuscript has been modified by a native English revisor.
Round 2
Reviewer 1 Report
I thank the authors for their response
Reviewer 2 Report
Thank you for making the suggested changes.